# Genetic Mechanism of Non-Targeted-Site Resistance to Diquat in *Spirodela polyrhiza*

**DOI:** 10.3390/plants13060845

**Published:** 2024-03-14

**Authors:** Martin Höfer, Martin Schäfer, Yangzi Wang, Samuel Wink, Shuqing Xu

**Affiliations:** 1Institute for Organismic and Molecular Evolution (iomE), Johannes Gutenberg University, 55128 Mainz, Germanymschaefer@uni-mainz.de (M.S.);; 2Institute for Evolution and Biodiversity, University of Münster, 48149 Münster, Germany

**Keywords:** non-targeted-site herbicide resistance, diquat, *Spirodela polyrhiza*, duckweed, GWAS, dose–response measurements

## Abstract

Understanding non-target-site resistance (NTSR) to herbicides represents a pressing challenge as NTSR is widespread in many weeds. Using giant duckweed (*Spirodela polyrhiza*) as a model, we systematically investigated genetic and molecular mechanisms of diquat resistance, which can only be achieved via NTSR. Quantifying the diquat resistance of 138 genotypes, we revealed an 8.5-fold difference in resistance levels between the most resistant and most susceptible genotypes. Further experiments suggested that diquat uptake and antioxidant-related processes jointly contributed to diquat resistance in *S. polyrhiza*. Using a genome-wide association approach, we identified several candidate genes, including a homolog of dienelactone hydrolase, that are associated with diquat resistance in *S. polyrhiza*. Together, these results provide new insights into the mechanisms and evolution of NTSR in plants.

## 1. Introduction

More than 500 unique cases of herbicide resistance have been discovered worldwide [1]. Many of these herbicide resistance cases are caused by genetic changes other than the target site and are therefore referred as non-targeted-site resistance (NTSR) cases [2]. Since many NTSR cases constitute complex and polygenic traits, our current understanding of the evolution and mechanism of NTSR remains limited.

Due to their lack of target site resistance mechanisms, bipyridinium herbicides, such as diquat and paraquat, constitute an ideal system for deciphering the genetic basis of non-target-site resistance mechanisms. These herbicides act by diverting electrons from the photosystem I (PSI), thereby competing with ferredoxin for its binding place on PSI [3,4]. Upon electron acceptation from PSI, they transfer one electron to molecular oxygen, generating superoxide anions. The reactive oxygen species (ROS) produced by this Mehler-like reaction lead to the destruction of thylakoid membranes and a decrease in the photosynthesis rate of the plant, which ultimately causes cell death by the induction of apoptosis [5]. With the widespread usage of bipyridinium herbicides for weed and algae control in agriculture and aquaculture [6,7,8,9], numerous resistance cases have been reported since the 1980s [10,11]. Most of these resistance cases exhibited a cross-resistance between paraquat and diquat [6,10,12,13,14]. From a mechanistic point of view, the vast majority of these cases were due to the vacuolar sequestration/chloroplast exclusion of the herbicide [13,15,16,17,18,19], an increased activity of the plant’s antioxidant radical scavenging system [20,21], or an intracellular-uptake-associated mechanism [6,22].

Most of the known herbicide resistance mechanisms are also shown to be involved in stress responses to environmental factors such as salinity [23,24], oxidative stress [24], heavy metal exposure [25], and cold stress [24]. Most prominently, ABC transporters, a class of proteins involved in various stress responses, such as those to drought [26], exposure to heavy metals [27], and pathogens [28], are involved in paraquat resistance [25,29]. These findings suggest that NTSR might have a pleiotropic origin from stress response or secondary metabolite pathways, thus explaining the short evolutionary time scale between the commercialization of herbicides and the appearance of resistant weed species. Yet, the underlying genetic principles remain poorly understood for many cases of bipyridinium herbicide resistance.

Here, we used giant duckweed *Spirodela polyrhiza* (L.) Schleid. (Arales: Lemnaceae) as a model system to study the mechanisms of diquat resistance. Because of their high susceptibility towards bipyridinium herbicides [30], their small body, and fast clonal reproduction [31], duckweeds are often used for toxicological studies on herbicides. Among duckweeds, *S. polyrhiza* has the smallest genome size and low genetic variation [32,33]. The available large genetic diversity panel and re-sequencing data on *S. polyrhiza* will also enable the identification of genetic mechanisms using a genome-wide association (GWA) approach. In this study, we focused on two main questions: (1) To what extent do *S. polyrhiza* genotypes vary in terms of their diquat resistance levels? (2) What are the underlying molecular and genetic mechanisms of diquat resistance?

## 2. Results

### 2.1. Intraspecific Variations in Diquat Resistance in S. polyrhiza

As the intra-specific variations in resistance are dependent on the applied diquat concentration, we used a stepwise approach to select the diquat concentration for screening. First, we used two randomly selected genotypes to perform a dose–response curve in *S. polyrhiza*. The results suggested that the EC10 and EC90 values for inhibiting FW (fresh weight) biomass in *S. polyrhiza* are ~1.8 nM and ~11.1 nM, respectively. Measuring the inhibition of the relative growth rate of the frond number, we obtained EC10 and EC90 values of ~2.2 nM and ~53.0 nM, respectively (Figure A1, Table A1). We then screened diquat resistance among 19 genotypes using 5 and 10 nM of diquat. We found the broad-sense heritability of diquat resistance to be higher at 10 nM (H2 = 0.98) than at 5 nM (H2 = 0.90). Furthermore, several genotypes showed hormetic effects at 5 nM of diquat concentration, whereas at 10 nM, we found a clear inhibition ranging from 22% to 84% across all genotypes (Figure A2). Therefore, we chose 10 nM as the concentration for screening diquat resistance among 138 genotypes.

Among the screened genotypes, diquat resistance varied significantly, either estimated based on the RGR (relative growth rate) of the frond number, the RGR of the frond area, or the decrease in biomass (FW) (Figure A3). Of all the quantified parameters, the RGR of the frond number was the most reproducible for quantifying diquat resistance (Figure A4). We did not detect significant differences in diquat resistance between the four genetic populations of *S. polyrhiza* (Tukeys HSD) identified using genome-wide markers (Figure A5). To further validate the variation in diquat resistance in our genotype accession, we performed dose-response experiments and estimated the EC50 values of the eight most resistant and eight most susceptible genotypes identified in the screening experiment (Table A2). The estimated EC50 value of the most resistant genotype (SP050, registered four-digit code: 0090) is 8.5-fold higher than that of the most susceptible genotype (SP011, registered four-digit code: 9242) (Figure 1 and Table A2). The differences in diquat resistance between SP050 and SP011 are also reflected by the decrease in two chlorophyll fluorescence parameters: Fv/Fm and Fv’/Fm’, in response to diquat. During the first 8 h of diquat treatment, SP050 has a much slower decay of Fv/Fm and Fv’/Fm’ than SP011 (Figure 1C,D). Together, these results suggest that diquat resistance greatly varied among different genotypes of *S. polyrhiza*.

### 2.2. Changes in Diquat Uptake and Antioxidant Capacity Contribute to Diquat Resistance

We compared the diquat uptake and antioxidant capacities among different genotypes to understand the molecular mechanisms of variations in diquat resistance. We found that diquat concentration in whole plants significantly correlates with inhibition effects, suggesting that reduced diquat uptake capacity increases diquat resistance (Figure 2A). However, the diquat concentration in frond tissue might be lower in highly resistant genotypes due to their larger biomass accumulation compared to susceptible genotypes after the 7 days of bioassay. To address this issue, we further compared the diquat uptake kinetics between SP050 and SP011 within 48 h, during which diquat did not affect plant biomass considerably. The results showed that SP011 accumulated significantly more diquat than SP050 after 24 h (Figure 2B), indicating their differences in diquat uptake capacity. A comparison of the diquat concentration between root and frond tissue suggested that roots have an approximately three-fold higher diquat concentration than fronds (Figure 2C). However, differences in diquat uptake between SP011 and SP050 were likely not due to their differences in root/frond diquat concentration ratio, which are similar between the two genotypes (Figure 2C). A measurement of the medium concentration of diquat did not reveal large fluctuations in cultures of SP050 and SP011 within a period of 7 days (Table A3). Due to the high stability of diquat in plant tissue and medium samples, the measured uptake kinetics are unlikely to be confounded by degradation processes.

We then measured the antioxidant capacity by quantifying the glutathione and ascorbate content levels among eight *S. polyrhiza* genotypes. Glutathione and ascorbate play a central function in protecting the plant from oxidative stress via the Asada–Halliwell–Foyer cycle (Figure 3A), which controls the redox hub by detoxifying superoxide anions generated by diquat. The quenching process of ROS occurs through the synergistic action of several enzymes, including the superoxide dismutase (SOD) via the conversion of the reductive metabolites AsA and GSH to their oxidized counterparts (Figure 3A), dehydroascorbate (DHA) and oxidized glutathione (GSSG), respectively. Therefore, we used the activity of SOD together with the ratios of AsA/DHA and GSH/GSSG as proxies to estimate antioxidant stress responses to diquat [23,24,34,35]. Diquat exposure decreased the GSH/GSSG and AsA/DHA levels in SP011 but not in SP050 (Figure 3B,C). For SP011, both ratios reached a minimum after 4 h of diquat treatment, followed by a recovery to their initial value after 24 h. The decline in the ratios was mainly reflected by a decrease in GSH and AsA concentrations in plant tissue (Figure A6). Additionally, we found a significant correlation between diquat tissue concentration and GSH/GSSG levels (Figure A7A) but not AsA/DHA levels (Figure A7B). However, the GSH/GSSG and AsA/DHA values were not correlated with resistance levels (Figure A7C,D). The enhanced maintenance of antioxidant ratios in SP050 is also reflected by the genotype’s elevated SOD activity under diquat treatment over SP011 (Figure 3D). Together, these data suggest that SOD activity under diquat treatment is correlated with resistance levels to diquat (Figure A8A). However, no such association was observed for the constitutive SOD activity (Figure A8B).

### 2.3. Genetic Basis of Diquat Resistance in S. polyrhiza

To identify the genetic basis of diquat resistance, we applied a GWA approach using both the single-nucleotide polymorphisms (SNPs) and structure variations (SVs) that were identified in our previous study [37]. Among all 42,462 SNPs, only one synonymous SNP (C to T) located within the coding sequence of the guanylate-binding-protein-like 2 *SpGBPL2* (SpGA2022_050909) showed a significant association with diquat resistance quantified based on FW only (Figure A9 and Figure A10, Table A4). Among the 842 SVs, we found that a 94 bp deletion located in the seventh intron of the mitochondrial sulfur dioxygenase *SpETHE1* (SpGA2022_053619) is associated with diquat resistance quantified based on all growth parameters (Figure 4A–C and Figure A11, Table A5). Genotypes that are homozygous at the deletion locus were significantly more resistant to diquat than heterozygous genotypes or genotypes lacking the deletion. Also, heterozygous genotypes were on average more resistant than genotypes lacking the deletion (Figure 4D). In addition, we found a 184 bp deletion within the fourth exon and the fourth intron of the lipoxygenase 2.1 *SpLOX2.1* (SpGA2022_008887) (Figure 4B, Figure A11 and Figure A12, Table A5) and a 56-bp deletion in an intergenic region (Figure 4C and Figure A11, Table A5) that is ~60 Kb far from the next open reading frame to be associated with diquat resistance. However, the 184 bp and 56 bp deletions are specific to only one of the growth parameters that were used for quantifying resistance effects.

To further validate the association between the 94 bp deletion in the *SpETHE1* intron and diquat resistance, we amplified the deletion region using PCR in the genomes SP050 and SP011. These two genotypes showed different levels of diquat resistance. The results confirmed that SP050 is homozygous for the identified deletion and that SP011 is lacking the deletion completely (Figure A13A). We then aimed to examine whether the deletion affected the expression or splicing of *SpETHE1* using qPCR. To this end, we conducted a time-course experiment and analyzed the expression and splicing of SpETHE1 in the roots and fronds, respectively. We could not find any differences in the splicing pattern between SP050 and SP011, neither for the control nor for the diquat-treated samples (Figure A13B). We found that *SpETHE1* expression was elevated in the root tissue relative to the frond tissue, approximately by a factor of three in both genotypes (Figure A14B). However, under control conditions and within 4 h of diquat treatment, the expression levels were similar between SP011 and SP050 (Figure 5 and Figure A14). After 72 h of diquat treatment, in the fronds, *SpETHE1* expression was 21% lower in SP050 compared to SP011, and it showed similar expression in the roots (Figure 5A and Figure A14A). The differential expression is likely due to the differences in diquat uptake between the two genotypes.

Because the genome of *S. polyrhiza* is compact and cis-regulatory elements can be located further away from the gene, we examined the expression of four genes located near *SpETHE1*: *SpDLH* (SpGA2022_012161), *SpALA2* (SpGA2022_053617), *SpSBE3* (SpGA2022_053621), and *SpTRB1* (SpGA2022_012163) (Figure 5B–F and Figure A14). Among them, the expression of *SpDLH*, the start codon of which is located 9.6 Kb downstream of the deletion (Figure 5B), is ~67% lower in the root tissue of SP050 compared to SP011 (Figure A14C). In the frond tissue, the expression was ~45% lower in SP050 compared to SP011 (Figure 5C). *SpDLH* is a homolog of dienelactone hydrolase, which is involved in the degradation of xenobiotics such as chlorocatechol [38] and might be linked to apoptotic processes as well [39]. In addition, the expression of *SpSBE3*, the start codon of which is located 7.7 Kb from the deletion, is 21% lower in the frond tissue of SP050 than in SP011 (Figure 5C). The expression of the other two genes was not different between SP050 and SP011.

## 3. Discussion

Here, we investigated the diquat resistance in *S. polyrhiza* using the worldwide diversity panel and identified gene candidates associated with the resistance. We found that the levels of resistance varied among genotypes by a factor of 8.5, likely due to the combination of changes in diquat uptake kinetics and antioxidant capacity. We used a GWAS approach to show that several candidate genes involved in metabolic processes or stress responses to biotic or abiotic factors are likely associated with diquat resistance.

Our results did not reveal any gene candidates with a transporter or carrier function that might have caused the observed differences in diquat uptake. Yet, since bipyridinium herbicides were also shown to bind to plant cell walls, restricted herbicide movement into the apoplast might not be exclusively explained through the plasma membrane transporter but also by the genotype’s cell wall composition as shown in paraquat-resistant biotypes of *Hordeum glaucum* (L.) Steud. (Poales: Poaceae) and in *Rehmannia glutinosa* (Gaertn.) Steud. (Lamiales: Orobanchaceae) [13,40]. Since we did not check for uptake differences on apoplast and protoplast levels separately, we cannot rule out the involvement of such extracellular movement barriers in diquat uptake. On a systemic level, increased retention of diquat in root tissue seems unlikely to explain differences in resistance levels since the root/frond ratio of the diquat concentration was similar across resistance levels. Alternatively, the observed difference in the diquat concentrations can be due to different capacities in metabolizing the herbicide.

From a physiological point of view, the increased diquat resistance of SP050 was characterized by an increased SOD activity and stable antioxidant ratios under diquat exposure compared to susceptible genotypes. Such genotypic-specific changes in AsA/DHA ratios were previously related to differences in SOD activity [35]. Since the GSH/GSSG ratio and diquat tissue concentration were significantly correlated after 24h of diquat exposure, some of the observed differences in antioxidant ratios might be influenced by diquat uptake kinetics as well. This suggests that diquat resistance in *S. polyrhiza* is likely facilitated by a synergistic involvement of uptake-related and antioxidant processes. Related to this, it was proposed that antioxidant processes only provide short-term protection against herbicide-mediated oxidative stress before more potent mechanisms such as sequestration or reduced uptake of the herbicide secure the long-term survival of resistant genotypes [41], which might explain why diquat-caused changes in the antioxidant ratios were only transient across different resistance levels.

From a genetic perspective, a 94-bp intronic deletion in *SpETHE1* was the only marker associated with several resistance parameters. Since we did not find significant differences in gene expression or splicing patterns associated with the deletion, the direct functional involvement of *SpETHE1* in diquat resistance seems unlikely. Yet, the differential expression of *SpDLH* and *SpSBE3*, two genes downstream of *SpETHE1*, as a potential consequence of the deletion, might suggest the involvement of long-distance gene regulation mechanisms. Such regulatory mechanisms seem to be more common in plant species with large or middle-sized genomes, such as *Zea mays* L. (Poales: Poaceae) [42,43] or *Oryza sativa* L. (Poales: Poaceae) [44] than in species with small genomes like *Arabidopsis thaliana* (L.) Heynh. (Brassicales: Brassicaceae) [45,46]. Yet, enhancement of transcription through chromatin looping was also shown to be established by introns [47], which might be a process more common in species with compact genomes that lack long intergenic sequences like *S. polyrhiza*.

Many homologs of our candidate genes were involved in various metabolic or stress response pathways. Whereas GBPLs and lipoxygenase like *SpLOX2.1* are involved in response mechanisms to pathogen infection [48,49,50], *SBE3* is involved in starch metabolism [51]. *DLH* functions in xenobiotic degradation [38,52] and apoptosis [39]. Their association with diquat tolerance suggests a pleiotropic function in herbicide resistance for these genes.

The conceptual connection of pleiotropic gene function to bipyridinium herbicide resistance was first established in *Conyza canadensis* (L.) Cronquist. (Asterales: Asteraceae) [53], where resistance to paraquat was inherited by a single gene that controls the regulation of multiple antioxidant enzymes [53]. In *Avena fatua* L. (Poales: Poaceae), multiple genes with pleiotropic functions that were involved in xenobiotic catabolism, redox maintenance, secondary metabolite pathways, and stress response pathways were differentially expressed at constitutive levels between resistant and susceptible biotypes [54]. These findings suggest that resistant genotypes benefit from a faster upregulation of their stress response when they experience herbicide exposure [54,55]. Since herbicide treatments trigger a systemic stress response, it has been argued that sub-lethal doses of herbicides will select genotypes with a constitutive upregulation of stress response-associated genes [55]. Yet, since many of our genotypes have likely never experienced direct selection through sub-lethal doses of diquat, this concept is only partially suitable for explaining the observed variation in diquat resistance. Alternatively, non-targeted-site resistance to herbicides can also be induced by heterogenous environments [56], suggesting that the observed resistance to diquat in *S. polyrhiza* also might have evolved from a simultaneous adaption to multiple environmental stress factors [57].

Taken together, this study paved the foundation for understanding the evolution of diquat resistance in *S. polyrhiza*. Further studies validating the function and mechanisms of the identified candidate genes will provide further insights into the evolution of herbicide resistance in plants and help develop more sustainable weed management strategies.

## 4. Materials and Methods

### 4.1. Plant Material and Cultivation Procedure

For pre-cultivation, we propagated plant material of *S. polyrhiza* in Erlenmeyer flasks for 14 days before each assay. For all assays and pre-cultivations we grew our *S. polyrhiza* genotypes in N-medium [58] at 26 °C, with 135 µmol photons·m^−2^·s^−1^ of light per day, a light/dark rhythm of 16h/8h and 50% humidity (GroBanks, model BB-XXL.3+cLED, CLF Plant Climatics, Wertingen, Germany) if not stated otherwise. Before each assay, we sterilized all of the analyzed genotypes via a combined protocol involving Klorix and Cefotaxime [59]. For all of the conducted assays, we grew the plants in plastic beakers (Verpackungsbecher PP, transparent, round, 250 mL, Plastikbecher.de GmbH, Giengen, Germany) filled with 150 mL N-medium (control conditions). We covered the cultures using perforated lids to allow air exchange. For the herbicide treatments, we supplemented the N-medium with the indicated concentration of diquat (Diquat-dibromide monohydrate, CAS:6385-62-2, Supelco, St. Louis, MO, USA). For all diquat measurements, the herbicide concentration is always expressed as the concentration of diquat-dibromide. Diquat was always applied as an aliquot from aqueous stocks. At the starting point of each experiment, we added fronds as colonies to the medium. If not stated otherwise, we triplicated each condition and genotype in all experiments.

### 4.2. Quantification of Diquat Toxicity

We expressed the diquat resistance of each genotype based on the changes in the fitness parameter relative growth rate (RGR) of frond number and frond area as well as the inhibition of fresh weight (FW) relative to an untreated control. To estimate the first two parameters, we counted the frond number and determined the surface area overgrown with fronds for each culture at the beginning and end of each assay. To this end, we took a picture of each culture at the beginning and end of each toxicity assay. To quantify the frond area, we added a reference stone with a black square of an edge size of 1 cm to each culture. At the endpoint of the cultivation, we harvested the plant material, dried it briefly with tissue paper, and stored it in pre-scaled reaction tubes to determine the FW. The procedure mentioned above applies to all toxicity assays if not stated otherwise. Based on these data, we calculated the RGR using a published method [31]:

$RGR=\frac{lnNE-lnN0}{t}\;\;$ where *NE* is the frond number/area at the endpoint of the toxicity test and *N*0 is the frond number/area at the starting point of the toxicity test.

We expressed the inhibition of the herbicide on the growth of the genotypes as an inhibitory effect (*IE*) defined by the following formula for all fitness parameters mentioned above:

$IE=\frac{M-X}{M}$ × 100 where *M* is the mean of an untreated control and *X* is the mean of the treatment.

#### 4.2.1. Determination of Screening Concentration

First, we aimed to identify a suitable diquat concentration that affects the growth in most of the genotypes. To make a rough estimate, we applied a diquat concentration range of 2 nM to 500 nM to two randomly selected genotypes, SP035 (registered four-digit code: 0109) and SP077 (registered four-digit code: 9508). For each replicate, we inoculated ten fronds at the start of cultivation and harvested the plant material after 10 days of cultivation. For each concentration, we quantified the growth inhibition as the IE of the FW and the RGR of the frond number. To estimate a screening concentration, we calculated the EC10 and EC90 values for both parameters and genotypes using the EC function from the “drc” R-package [60,61]. Based on the information obtained from this experiment, we further narrowed down the range of potential screening concentrations by testing the inhibition of 19 genotypes to 5 nM and 10 nM of diquat. Our criterion for choosing a suitable screening concentration was that no hormetic effects were visible for any of the 19 genotypes. To avoid the overgrowth of cultures, we decreased the cultivation time to 7 days and inoculated only six fronds at the starting point of the experiment. We quantified diquat resistance based on the IE of the RGR frond number (IE of RGR frond number), the IE of the RGR frond area (IE of RGR frond area) and the IE of the fresh weight (IE of FW). We further estimated the broad-sense heritability of diquat resistance at 5 and 10 nM according to a published method [62] using the H2cal function of the inti R-package [63].

#### 4.2.2. Diquat Resistance Screening

We assayed 138 genotypes in five batches, each encompassing 23 to 35 genotypes, regarding their growth inhibition at a diquat concentration of 10 nM for 7 days. A list of all screened genotypes is provided in the Appendix A. At the beginning of the cultivation, we inoculated six fronds in each culture. To control for batch effects, we cultivated genotype SP028, which continuously served as an internal standard throughout all batches. We then normalized the inhibition of all genotypes to that of SP028 in the respective batch. We used these normalized the values of the IE of RGR the frond number (norm. IE RGR of frond number) and area (norm. IE of RGR frond area) and the IE of the FW (norm. IE of FW) to identify the most resistant and susceptible genotypes.

#### 4.2.3. Dose–Response Measurements

To quantify the variation in diquat resistance levels in our 138-genotype accession, we estimated the half maximal effective concentration (EC50) values for the eight most resistant and the eight most susceptible genotypes identified in our resistance screening. We applied a concentration range from 0.5 nM to 100 nM of diquat to a population of six fronds at the starting point of the assay. The duration of the toxicity test was 8 days. To correct for the occurrence of hormetic effects, we calculated all dose–response curves using the drc R-package, applying the hormetic Cedergreen–Ritz–Streibig model UCRS.4c, which has previously been used in toxicity tests of herbicide formulations on duckweed species [64]. In contrast to normal logistic models, which are strictly monotonous functions, the UCRS4.c model takes stimulatory effects, that are caused by hormesis into account [64]. Therefore, the UCRS.4c model estimates the potency of the herbicide via a lower bound on the ED50 value [64]. Since the hormetic effects were minimal in all cases, the lower bound on ED50 provides a close approximation of the EC50. Therefore, we refer to the lower bound on ED50 as an estimate of EC50 in this manuscript.

### 4.3. Chlorophyll Fluorescence Measurement

We evaluated diquat resistance on a physiological level, measuring the decay of the chlorophyll fluorescence parameters Fv/Fm and Fv’/Fm’ in response to high levels of diquat. The Fv/Fm ratio indicates the decrease in the quantum yield of linear electron flow through PSII [65] and the Fv’/Fm’ ratio is an estimate of the maximum quantum yield of PSII in a light-adapted state [66]. We quantified Fv/Fm and Fv’/Fm’ for the previously identified four most resistant (R) and four most susceptible genotypes (S). For this, we acclimated all of the plant cultures at 26 °C at 110 µmol photons·m^−2^·s^−1^ of light per day and a light/dark rhythm of 16 h/8 h (Percival, model AR-41L3, CLF Plant Climatics) for two days before measurement. We used 12-well plates (Nunc, Roskilde, Denmark) filled with 6 mL of N-medium containing 40 µM of diquat for herbicide exposure of the plant material. Each well was inoculated with a duckweed colony composed of three to four fronds. Each genotype was analyzed with three biological replicates for each time point. For each replicate, three areas of interest were defined. We chose durations of 0 min, 30 min, 1 h, 2 h, 4 h, 8 h, 12 h, and 24 h as endpoints for the incubation. We conducted fluorescence measurements using an I-PAM system (M-series, MAXI version, Heinz Walz GmbH, Effeltrich, Germany), applying all settings published previously for chlorophyll fluorescence measurements in *Lemna minor* L. (Arales: Lemnaceae) [67]. For dark adaption, we incubated our samples in the dark for ten min before the saturation pulse. After recording Fv/Fm, we adapted the plants to actinic light at 84 PAR (photosynthetically active radiation) for 10 min, before measuring the Fv’/Fm’ ratio.

### 4.4. Superoxide Dismutase Activity

We measured SOD activity for the four most susceptible (S) and four most resistant (R) genotypes identified in our dose–response experiment. To determine the superoxide dismutase (SOD) activity from the plant material, we cultivated five colonies of each genotype under 10 nM of diquat or under control conditions for 24 h. Next, we suspended 10 mg (FW) of plant material from each sample in 500 µL of 0.01 M MOPS buffer (Carl Roth, Karlsruhe, Germany) at pH 7.2. We homogenized the suspension by pulsed mixing for 5 s and centrifuged each sample at 14,000× *g* for 1 min before analysis. For the quantification of the SOD activity, we diluted 150 µL of supernatant 1:4 in MOPS buffer pH 7.2. We measured SOD activity following the instructions of the SOD determination kit (Product 19160, SOD Assay Kit, Sigma-Aldrich, St. Louis, MO, USA). To determine the protein concentration, we diluted the supernatant 1:2 in MOPS buffer at pH 7.2 and analyzed the samples according to the manufacturer’s instructions using the Rapid Gold BCA Protein Assay Kit (Thermo Fisher Scientific, Waltham, MA, USA). We standardized SOD activity to the measured protein content. To check for the involvement of SOD activity in diquat resistance, we further correlated the SOD activity with our estimates of EC50 values.

### 4.5. Ascorbate and Glutathione Measurements

To measure the impact of diquat exposure on shifts in the concentration ratios of ascorbate (AsA) to dehydroascorbate (DHA) and reduced (GSH) to oxidized glutathione (GSSG), we cultivated 30 fronds from each R and S genotype for 4, 8 and 24 h under 10 nM of diquat. We grew the control samples for 24 h in N-medium without diquat. For each genotype and measurement point, we made five biological replicates. At the endpoint of the assay, we harvested all of the plants, dried them with tissue paper, transferred them into pre-scaled reaction tubes, and froze them immediately in liquid nitrogen. We stored all of the samples at −80 °C until analysis. We extracted antioxidants from samples of 20 mg FW each with 250 µL of 5% (*w*/*v*) metaphosphoric acid (Carl Roth). The extracts were diluted 1:10 with 5% (*w*/*v*) metaphosphoric acid containing 1.11 µg/mL of isotope-labeled GSH (Glutathione-Glycine-^13^C_2_,^15^N, Sigma-Aldrich) (labGSH). The further steps of the extraction protocol are based on a method published previously with the modifications listed below [68].

We analyzed all of the antioxidants via LC-MS. For separation, we used a Nexera X3 UHPLC system (Shimadzu, Kyoto, Japan) equipped with a Nucleodur Sphinx RP column (250 × 4.6 mm, 5 µm, Macherey-Nagel, Düren, Germany) and a Nucleodur Sphinx RP EC 4/3 guard column (5 µm, Macherey-Nagel) maintained at 25 °C. For the mobile phase, we used an aqueous solution of 0.2% (*v*/*v*) formic acid, 0.05% (*v*/*v*) acetonitrile as buffer A, and acetonitrile as buffer B. For elution, we applied a gradient mode using the following program: 0.0 min/2% B, 3.5 min/2% B, 5.0 min/100% B, 8.0 min/100% B, 8.2 min/2% B, 12.5 min/2% B at a constant rate of 1 mL/min. The injection volume was 1 µL of extract per sample. For quantification we used an LC-MS/MS-8060 (Shimadzu) with an electrospray ionization source with the following parameters: nebulizing gas flow: 3 L/min, heating gas flow: 10 L/min, drying gas flow: 10 L/min, interface temperature: 350 °C, DL temperature: 250 °C, heat block temperature: 400 °C, CID gas flow: 270 kPa, interface voltage: 4000 V. GSH and labGSH were quantified in the positive ionization mode, all other metabolites were quantified in the negative ionization mode. We operated with a mass spectrometer in multiple reaction monitoring (MRM) mode using the following settings: Ch1: 308.09 → 76.1 *m*/*z* (CE −26 V), Ch2: 308.09 → 162.1 *m*/*z* (CE −16 V) for GSH, Ch1: 311.09 → 181.95 *m*/*z* (CE −11 V), Ch2: 311.09 → 165.2 *m*/*z* (CE −14 V) for labGSH, Ch1: 173.01 → 113.20 *m*/*z* (CE 10 V), Ch2: 173.01 → 143.05 *m*/*z* (CE 12 V) for DHA, Ch1: 175.02 → 86.85 *m*/*z* (CE 22 V), Ch2: 175.02 → 115.05 *m*/*z* (CE 13 V) for AsA and Ch1: 611.14 → 305.85 *m*/*z* (CE 24 V), Ch2: 611.14 → 271.90 *m*/*z* (CE 28 V) for GSSG. For all of the metabolites, Ch1 was used as the quantifier and channel Ch2 as the qualifier. We facilitated the quantification of GSH based on the internal standard labGSH. In the case of GSSG, AsA, and DHA, we quantified them using the internal standard and a conversion factor (CF) that was determined based on external standard curves (n_Analyte_ = n_labGSH_ × CF). The empirically determined CFs were 13.1 for AsA, 652.6 for DHA and 2.2 for GSSG.

### 4.6. Diquat Measurement

To check the possibility of an involvement of transport-related mechanisms in diquat resistance, we quantified the diquat concentration in plant tissue for all R and S genotypes. To relate diquat resistance to tissue concentration, we extracted diquat from freeze-dried samples of 138 genotypes. We pooled all triplicates of the same genotype treated with diquat. Due to a sample being lost during diquat extraction, the total number of samples was reduced to 137. The extraction procedure follows a previously described method with minor modifications [69]. We homogenized the freeze-dried plant material (DW) in a tissue lyser (Qiagen, Venlo, The Netherlands) by adding two to three metal beads and shaking for 1 min at 25 Hz. We aliquoted 10 mg of the ground plant material in 96-well biotubes (catalog number: MTS-11-C, MTS-11-C-R, Axygen, New York, NY, USA) and resuspended each sample with 800 µL of 40% (*v*/*v*) methanol, that was acidified with 0.0375 N HCl and contained 20 ng [^2^H_4_]-diquat dibromide (03627-5 mg, Supelco) as an internal standard. We homogenized the samples in a tissue lyser (Qiagen) at 20 Hz for 10 min, followed by an incubation step at 80 °C for 15 min and a second homogenization step at 20 Hz for 10 min. After two centrifugation steps at 2000× *g* for 20 min at 4 °C, we transferred the supernatant to a PCR plate for analysis. Due to the adhesion of diquat to glass surfaces, we did not use glass tubes throughout the whole extraction process. Before the measurement, we performed another centrifugation step at 2000× *g* for 20 min at 4 °C.

Since tissue dilution effects caused by the higher propagation rates of resistant compared to susceptible genotypes in the diquat treatments might have confounded the correlation between diquat resistance and tissue concentration, we established an uptake kinetic for the R and S genotypes. We cultivated the genotypes in N-medium supplemented with 10 nM of diquat for 15 min, 30 min, 1 h, 2 h, 4 h, 8 h, 12 h, and 24 h. We triplicated each treatment, with each replicate containing 15 fronds growing in colonies. We extracted 2 mg of homogenized plant material (DW) applying the workflow described above but using only 200 µL of extraction buffer containing 1 ng of the [^2^H_4_]-diquat dibromide as the internal standard per sample.

We measured the diquat concentration in the root and frond tissue separately to look for tissue-specific translocation patterns. We cultivated the genotypes at 10 nM of diquat for 24 h. Each replicate consisted of 30 fronds growing as colonies. We dissected samples into the root and frond tissue for separate analysis. The extraction of each 2 mg of tissue per sample was performed as described in the previous paragraph.

To monitor changes in the diquat concentration of the medium, we cultivated R and S genotypes for 7 days at 10 nM of diquat. Each day, a medium sample of 400 µL was taken from each replicate and immediately frozen in liquid nitrogen. After lyophilization, we resuspended the precipitate in 400 µL extraction buffer containing 4 ng of [^2^H_4_]-diquat dibromide. We then homogenized 300 µL of the supernatant in a tissue lyser as mentioned above and continued with the extraction as described for plant tissue samples.

We analyzed the diquat via LC-MS. For separation, we used a Nexera X3 UHPLC system (Shimadzu) coupled to an LC-MS/MS-8060 with an Acclaim Trinity Q1 column (100 × 2.1 mm, 3 µm, Thermo Fisher Scientific) and a Trinity Q1 guard column (10 × 2.1 mm, 5 µm, Thermo Fisher Scientific) maintained at 30 °C. The injection volume of the sample was 0.5 µL for all measurements. For the mobile phase, we used a buffer containing 75% (*v*/*v*) acetonitrile and 25 mM ammonium acetate with a pH adjusted to 5.0 with acetic acid in an isocratic mode at a flow rate of 0.4 mL/min. For quantification, we used an LCMS-8060 (Shimadzu) with an electrospray ionization source with the following parameters: nebulizing gas flow: 3 L/min, heating gas flow: 10 L/min, drying gas flow: 10 L/min, interface temperature: 300 °C, DL temperature: 250 °C, heat block temperature: 400 °C, CID gas flow: 270 kPa, interface voltage: 1000 V. The mass spectrometer operated in multiple reaction monitoring (MRM) mode using the following settings: Ch1: 183.09 → 157.00 *m*/*z* (CE −20 V), Ch2: 183.09 → 130.10 *m*/*z* (CE −31 V) for diquat and Ch1: 186.11 → 158.00 *m*/*z* (CE −20 V), Ch2: 186.11 → 131.10 *m*/*z* (CE −33 V) for [^2^H_4_]-diquat. For this measurement channel, Ch1 was the quantifier and Ch2 the qualifier. To check for potential associations between the uptake of diquat or the diquat resistance and the antioxidant levels, we correlated the diquat tissue concentration after 24 h and the resistance levels with the GSH/GSSG and AsA/DHA levels.

### 4.7. GWAS

To reduce the false positive signals due to clonality, we selected all sequenced genotypes from our 138 genotype accession and grouped them into 98 genets according to a recently published method [70]. From each genet the genotype with the highest sequencing coverage was selected as representative genotype. All of the analyzed genotypes are listed in the Appendix A. We used the normalized IE of FW, frond number and frond area as input parameters for the GWAS. We conducted all GWAS on the 98 representative genotypes using an univariate linear mixed model [71] implemented in the vcf2gwas package [72]. We used structural variants (SVs) (>50 nucleotides) and SNPs as input for our GWAS analyses. All genotypic data for SNPs and SVs, as well as the annotation, were part of a recent publication [37] and can be found under https://github.com/Xu-lab-Evolution/Great_duckweed_popg (accessed on 9 January 2024).

We formatted the genotype SV dataset as the input for vcf2gwas according to the published protocol [73]. Before analysis, we pruned our SNP and SV datasets using the PLINK package, which is integrated in the vcf2gwas platform, applying a window size of 100 markers, a step size of 10 markers, and allowing a phased r^2^-threshold of 0.33. We removed all SNPs and SVs with a minor allele frequency (MAF) of less than 5%, resulting in 842 SVs and 42,462 SNPs.

To estimate differences in diquat resistance on a population level, we sorted all genotypes into four genetic populations based on previously published information [37].

### 4.8. RT-PCR Analysis of Splicing Pattern in SpETHE1

To validate the presence of the identified mutations in the most resistant (SP050) and the most susceptible genotype (SP011), we isolated genomic DNA from these *S. polyrhiza* genotypes using the DNeasy Plant mini kit (Qiagen) and genotyped them using primers 1_ETHE1_fwd and 2_ETHE1_rev (Table A6). The presence of the 94 bp deletion would yield a 705 bp product, whereas a wild-type sample would give a 799 bp product. The PCR program included an initial denaturation step at 95 °C for 3 min followed by 31 cycles of 95 °C—15 s, 60 °C—15 s, 72 °C—70 s with a single final elongation step at 72 °C for 5 min. We visualized the products on a 1% agarose gel.

To check whether *SpETHE1* (SpGA2022_053619) shows different splicing patterns between SP050 and SP011, we cultivated 30 fronds of each genotype for 4, 8, and 24 h at a 10 nM diquat concentration. For each genotype, 30 fronds each were cultivated for 24 h in a medium without diquat were used as control. For all RNA sample preparations, we extracted <20 mg of FW aliquots using the InnuPREP RNA mini kit (Analytik Jena, Jena, Germany) and checked RNA integrity on a 1% agarose gel afterwards. We performed all cDNA synthesis reactions following the RevertAid First Strand cDNA synthesis kit (Takara, Shiga, Japan) using random hexameric primers to ensure cDNA synthesis from low-stability RNAs lacking polyA-tails. Next, we amplified fragments from the *SpETHE1* cDNA samples with the primers 3_ETHE1_fwd and 2_ETHE1_rev (Table A6), that bind on the fourth and nineth exon. The primers are flanking the intronic sequence carrying the deletion. Correct splicing is indicated by a fragment size of 330 bp. We carried out an amplification from genomic DNA using the same primer pair as the control, expecting a product of 1848 bp. The time program for amplification from cDNA template included an initial denaturation step at 95 °C for 3 min followed by 30 cycles of 95 °C for 15 s, 60 °C for 15 s, 72 °C for 30 s with a single final elongation step at 72 °C for 5 min. For amplification from genomic DNA, the duration of the extension step at 72 °C was increased to 150 s. We separated the products on a 2% agarose gel. For all amplifications, we used the PrimeSTAR HS DNA polymerase (Takara).

### 4.9. RT- qPCR for Monitoring Gene Expression in Duckweed Tissue

We measured the expression differences in *SpETHE1* and genes in close distance to *SpETHE1* in SP050 and SP011. To analyze the diquat-induced differences in expression levels of *SpETHE1* in root and frond tissue separately, we cultivated 40 fronds from each genotype for 4 and 72 h at 10 nM of diquat. As a control, we cultivate 40 fronds from each genotype at N-medium without diquat for 24 h. We made five biological replicates for each genotype and treatment. Since *ETHE1* showed a potential circadian and diurnal regulation pattern [74,75], we always harvested the treated samples and controls at the same time of day. Root and frond tissue were collected from each culture separately. The root and frond tissues of the control cultures were also used to determine the constitutive expression levels of the flanking genes *SpDLH* (SpGA2022_012161), *SpTRB1* (SpGA2022_012163), *SpALA2* (SpGA2022_053617), and *SpSBE3* (SpGA2022_053621). We conducted all cDNA syntheses as described in the previous section but with oligo-dT primers instead of random hexameric primers. For all qPCR experiments, we quantified *SpETHE1* according to a previously published method [76] using the glycerin-aldehyde-3-phosphate dehydrogenase (*SpGAPDH*, SpGA2022_054082) and alpha elongation factor one (*SpaEF*, SpGA2022_005771) as reference genes. The primer sequences of the two reference genes were published previously [37].

We calculated the primer efficiencies based on a dilution series of cDNA templates (Table A7) and evaluated their specificity by loading the reaction mixtures on a 2% agarose gel. Before starting the RT-qPCR, we diluted all cDNA samples 1:100 in water. We carried out the RT-qPCR using a RotorGene Q system (Qiagen), using a master mix from the KAPPA SYBR FAST kit (Roche, Basel, Switzerland). For each gene and sample, we made three technical replicates. The time program for qPCR included an initial denaturation at 98 °C for 3 min followed by 40 cycles of 98 °C for 3 s of denaturation and 60 °C for 20 s of annealing/extension.

### 4.10. Software and Statistics

We performed all statistical analyses using R version 4.2.0. We analyzed all of the pictures used to quantify the frond areas and numbers using ImageJ (version 1.53g) running on the Fiji platform. We accomplished the measurement of chlorophyll fluorescence using ImagingWinGigE version 2.47. For all LCMS measurements, we recorded the signals using LabSolutions version 5.97. We integrated peaks using Lab Solutions Insight version 3.5 SP2. We used the FSA package of R [77] to calculate the standard errors. For the statistical analyses of differences in chlorophyll fluorescence parameters, we applied the lme4 package [78] and the multcomp R packages [79] to implement a linear mixed effect model and a Tukeys post hoc test for multiple comparisons. For all other experiments involving the most resistant and most susceptible genotype, we compared mean values using the Student’s *t*-test. We used the *F*-test to examine the significance of our regressions. We used one-way ANOVA followed by a Tukey’s HSD post hoc test for comparisons of multiple mean values. If the criterion of homogeneity of variances was violated, we used the Games Howell test to compare multiple mean values. To conduct our GWAS, we used version 0.8.7. of the vcf2gwas platform. To determine significant genetic markers with a Bonferroni-corrected *p* < 0.05, we used a Wald test. We converted genotype files into vcf format, which is the only accepted format in vcf2gwas, using the TASSEL platform version 5.0 [80].

## 5. Conclusions

The findings of this study provide insights into the genetic and physiological principles of non-targeted-site resistance to diquat in *S. polyrhiza*. Since the analyzed genotypes showed gradual variation in their diquat resistance that could be associated with herbicide uptake kinetics and antioxidant responses, we suggest a polygenic mechanism behind the observed phenotypic variation in resistance levels. The association of gene candidates involved in apoptosis, pathogen response, starch metabolism and xenobiotic degradation points to a pleiotropic origin of diquat resistance from stress response pathways. Since most of the analyzed genotypes have likely never experienced selection through diquat, we suggest that the detected intra-specific variation in diquat resistance might originate from an adaption to environmental stress factors such as drought, UV stress, and herbivory.

Taken together our results contribute to a better understanding of the rapid evolution of novel resistance cases around the world. Association studies on the genetic basis of herbicide resistance are required to identify novel genetic signatures which can be used for the marker-associated breeding of herbicide resistance crop species or for the development of molecular screening methods for resistant weed species.

## Figures and Tables

**Figure 1 plants-13-00845-f001:**
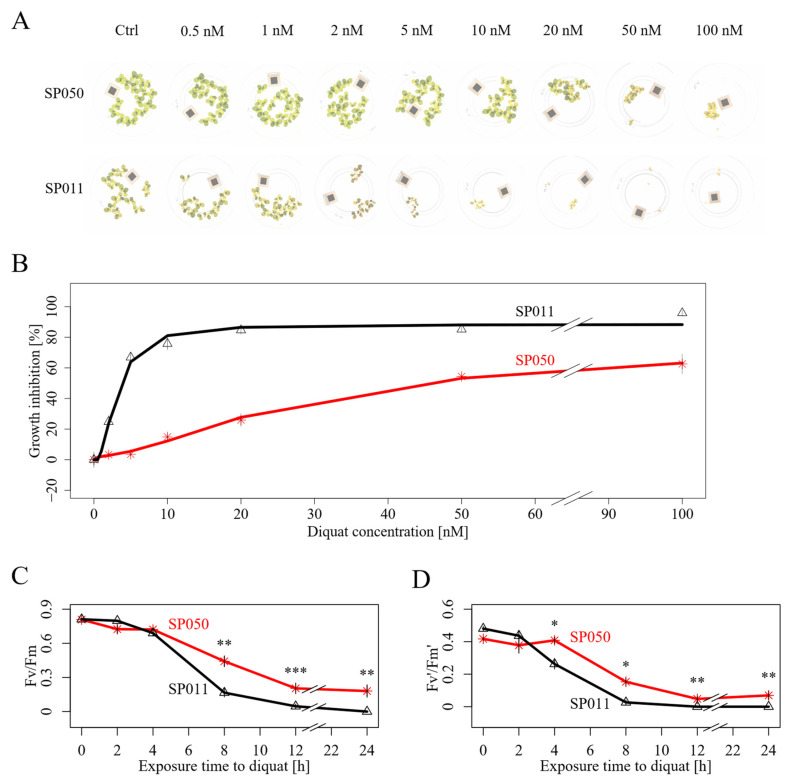
Comparison of growth kinetics of the most resistant genotype SP050 and the most susceptible genotype SP011 under diquat treatment (**A**) *S. polyrhiza* genotypes SP011 and SP050 grown under various diquat concentrations for 8 days. (**B**) Dose-response curves depicting the mean IE values with their corresponding standard errors calculated from three biological replicates for various diquat concentrations. We used the UCRS4c hormetic model from the drc R-package to estimate the EC50 values. The predicted functions used to derive this value were drawn as colored lines. (**C**,**D**) The development of the chlorophyll fluorescence parameters Fv/Fm (**C**) and Fv’/Fm’ (**D**) are shown over time. The data points constitute mean values from three biological replicates; error bars indicate ± standard error, and different significance levels were marked with asterisks: *—0.05 ≥ *p*-adjust > 0.01, **—0.01 ≥ *p*-adjust > 0.001, ***—*p*-adjust < 0.001 (linear mixed effect model, Tukeys HSD).

**Figure 2 plants-13-00845-f002:**
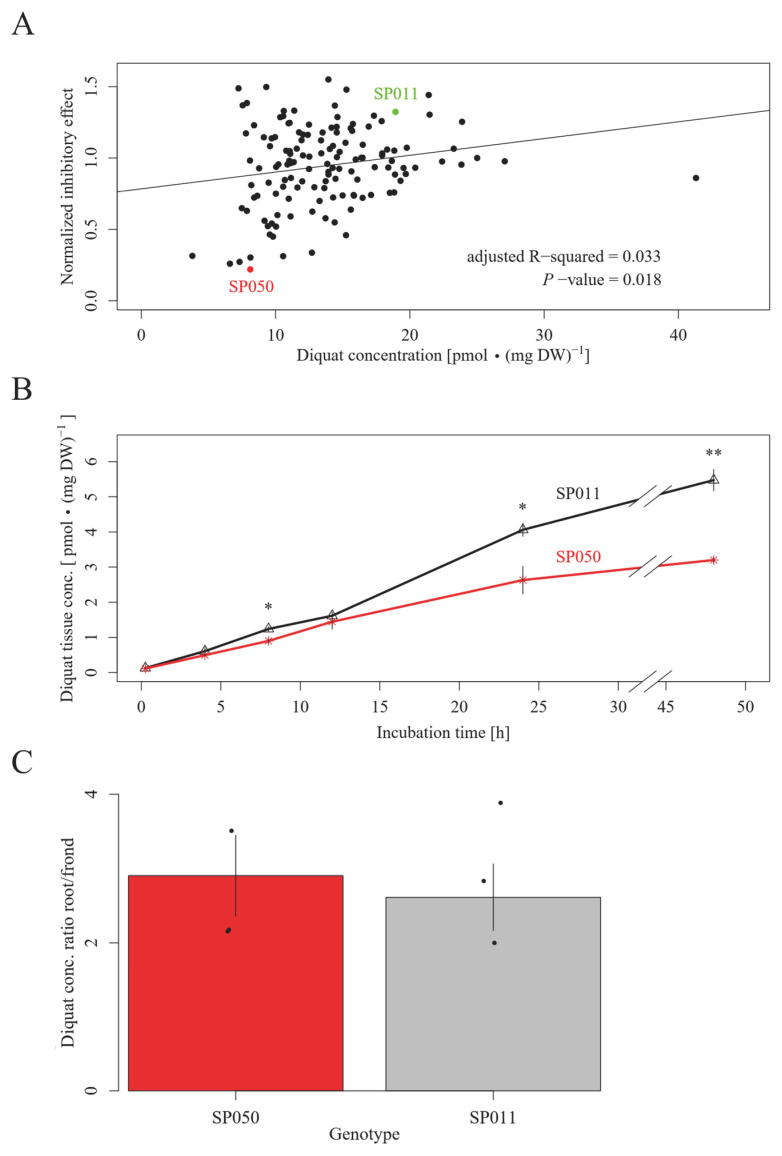
Uptake and translocation mechanisms of diquat. (**A**) The diquat resistance calculated based on the inhibition of the RGR of the frond number is significantly correlated (F-test) with the diquat tissue concentration of 137 genotypes. (**B**) After 24 and 48 h of diquat exposure, SP011 accumulated significantly more diquat than SP050. *—0.05 ≥ *p*-adjust > 0.01, **—0.01 ≥ *p*-adjust > 0.001 (two-sided Student’s *t*-test). (**C**) Both genotypes showed an approximately three-fold higher concentration of diquat in root tissue compared to frond tissue. No significant difference between the two genotypes was found (two-sided Student’s *t*-test). The diquat tissue concentration was measured after 24 h of diquat exposure.

**Figure 3 plants-13-00845-f003:**
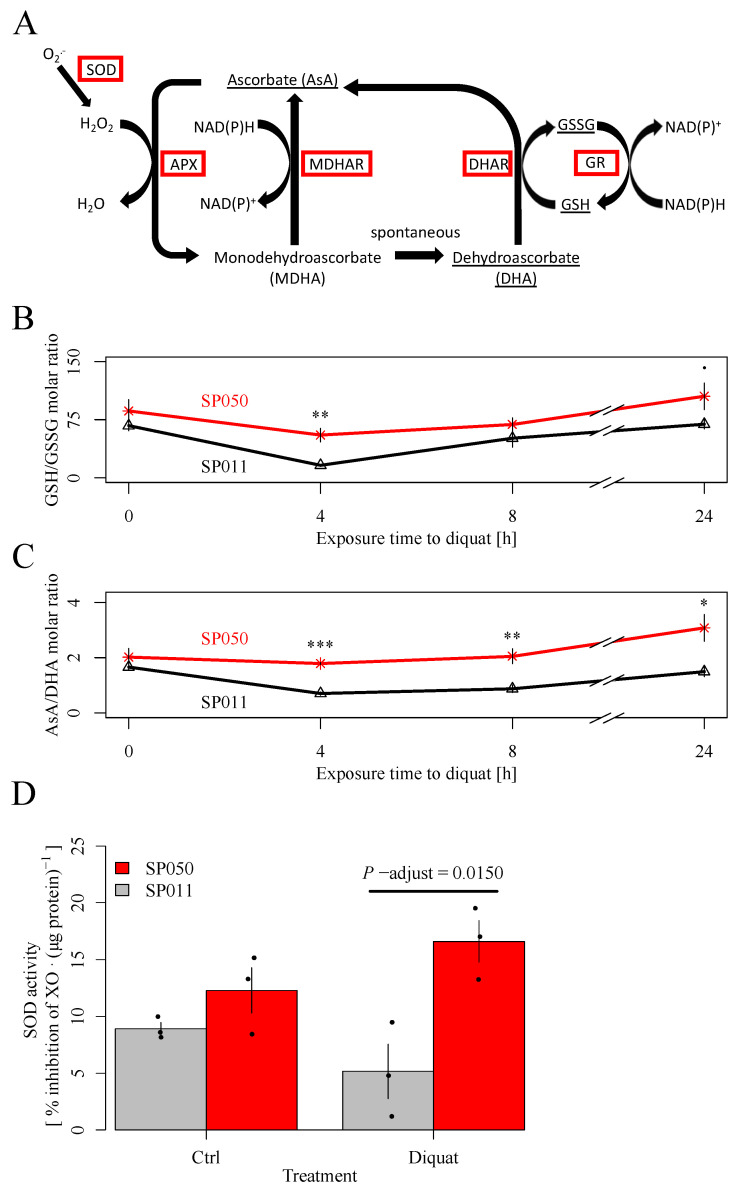
(**A**) Foyer–Asada–Halliwell cycle according to Noctor and Foyer, 1998 [36]. Enzymes are highlighted by red margins. Metabolites analyzed in this study are underlined. SOD converts superoxide anions to H_2_O_2_. Through oxidation of AsA, H_2_O_2_ is degraded to H_2_O by APX. AsA is regenerated by oxidation of GSH to GSSG. The latter is reconverted to GSH by GR, consuming NADPH. SOD—superoxide dismutase, APX—ascorbate peroxidase, MDHAR—monodehydroascorbate reductase, DHAR—dehydroascorbate reductase, GR—glutathione reductase. The time course of mean GSH/GSSG (**B**) and AsA/DHA (**C**) ratios of four to five replicates with their respective standard errors are shown for SP050 and SP011. SP050 exhibited a significantly increased GSH/GSSG ratio over SP011 after 4 h of diquat treatment and an increased AsA/DHA ratio after 4, 8, and 24 h of diquat treatment (unpaired two-sided Student’s *t*-test). (**D**) Mean SOD activity after 1-day incubation at 10 nM of diquat or control conditions for three biological replicates ± standard error. Genotype SP050 exhibited a significantly higher SOD activity than SP011 after 24 h of diquat exposure. *—0.05 ≥ *p*-adjust > 0.01, **—0.01 ≥ *p*-adjust > 0.001, ***—*p*-adjust < 0.001 (two-sided Student’s *t*-test).

**Figure 4 plants-13-00845-f004:**
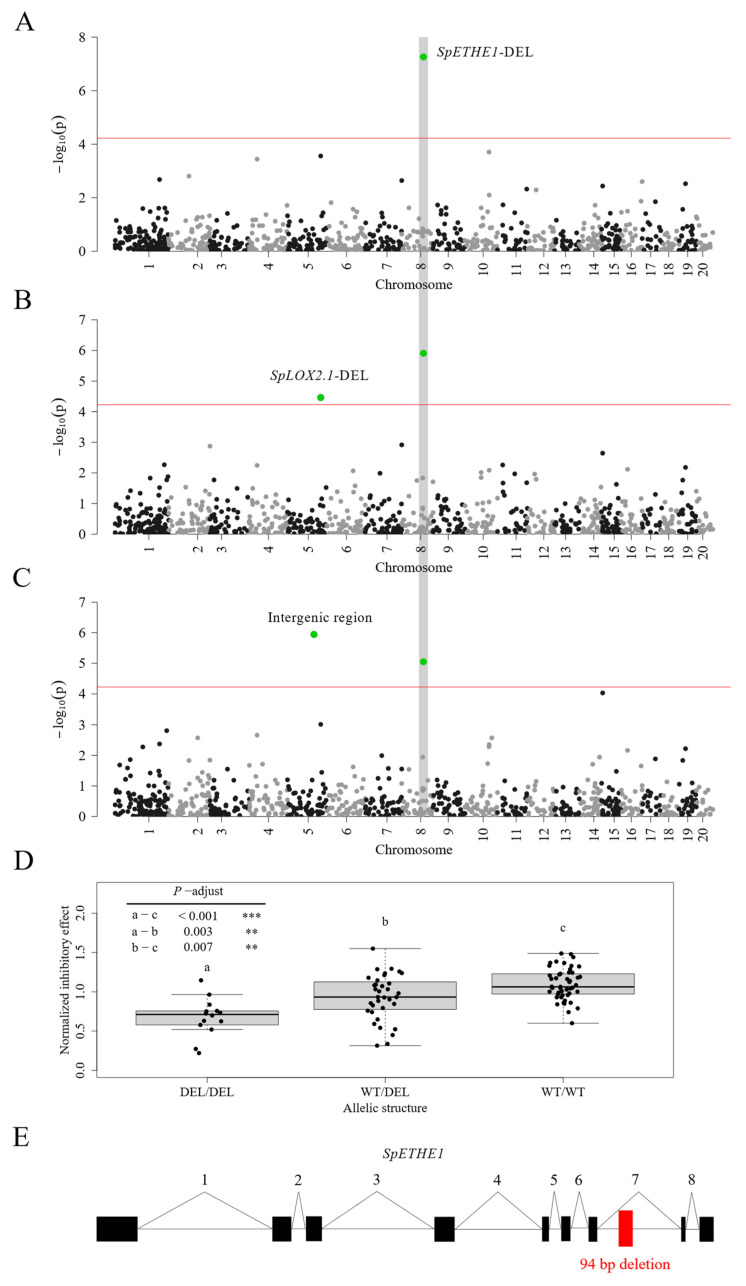
(**A**–**C**) Manhattan plots from a GWAS conducted on structure variations, highlighting the diquat resistance-associated 94 bp deletion in *SpETHE1*. All significant markers (Wald test) were highlighted as green dots. The Bonferroni corrected significance threshold (red line) at *p* < 0.05 is 5.94 × 10^−5^. The GWAS was conducted using normalized inhibition effects calculated based on frond number (**A**), frond area (**B**), and fresh weight (**C**). (**D**) Box plot of mean values of the norm. IE RGR of the frond number of genotypes carrying the respective alleles. On average, genotypes homozygous for the deletion were more resistant to diquat than heterozygous genotypes or genotypes without the deletion. **—0.01 ≥ *p*-adjust > 0.001, ***—*p*-adjust < 0.001 (Tukeys HSD). Heterozygous genotypes exhibited a greater mean resistance than genotypes without deletion. (**E**) Gene structure of *SpETHE1* with the 94 bp deletion in the seventh intron highlighted as a red box.

**Figure 5 plants-13-00845-f005:**
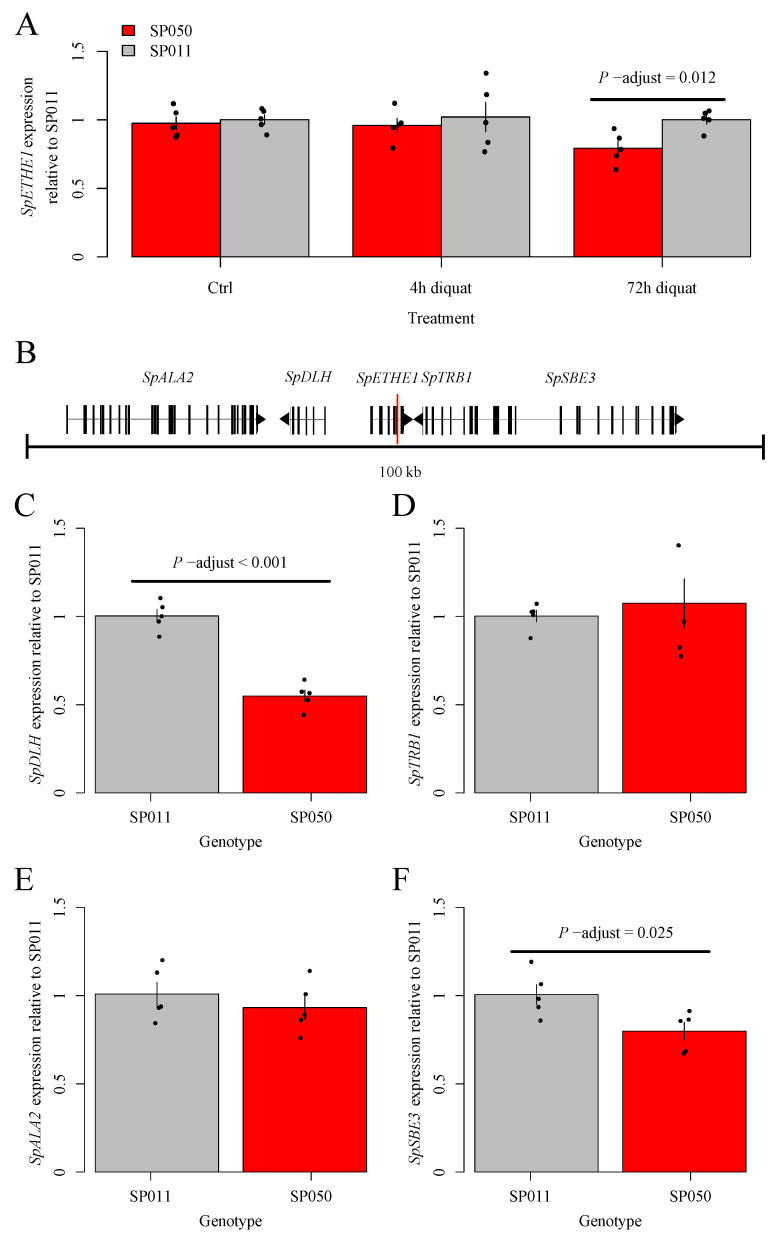
Relative transcript abundance of candidate genes in the frond tissue of *S. polyrhiza*. Mean values of five biological replicates are shown with their respective standard errors. (**A**) *SpETHE1* expression in SP050 and SP011 relative to its mean expression in SP011 without diquat treatment (Ctrl) or treated with diquat for 4 h and 72 h. After 72 h, *SpETHE1* was less expressed in SP050 than in SP011 (two-sided Student’s *t*-test). (**B**) The location of *SpDLH*, *SpTRB*, *SpALA*, *SpSBE3* and the deletion in *SpETHE1*. The deletion is depicted as a red line. Transcript abundance of *SpDLH* (**C**), *SpTRB1* (**D**), *SpALA2* (**E**), and *SpSBE3* (**F**) in SP050 relative to their mean expression in SP011. The transcript abundances of *SpDLH* and *SpSBE3* are lower in SP050 than in SP011 (two-sided Student’s *t*-test).

## Data Availability

Publicly available datasets including Appendix A were analyzed in this study. These data can be found here: https://doi.org/10.5061/dryad.2fqz612ww (accessed on 3 February 2024).

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
