# Peer review of "Genetic Mechanism of Non-Targeted-Site Resistance to Diquat in Spirodela polyrhiza"

_plants, 2024, doi:10.3390/plants13060845_

Round 1

Reviewer 1 Report

Comments and Suggestions for Authors

The manuscript “Genetic mechanism of non-targeted-site resistance to diquat in Spirodela polyrhiza”, by Höfer et al. (plants-2849712) presents an in-depth study of genetic resistance to diquat of different genotypes of a model organism, the giant duckweed Spirodela polyrhiza. The study is well conducted and comprehensive, and requires only some minor revisions.

The order in which figures and tables (including supplementary ones) are cited should be carefully checked. References at the end of the manuscript should also be checked for compliance with the journal style.

Other suggestions are enlisted below.

Line 54. Please add “(L.) Schleid. (Arales: Lemnaceae) after “Spirodela polyrhiza”. Accordingly, authority and classification should be indicated every time first a species or a genus is cited for the first time in the text.

Lines 73, 75, 77, 357, 358, 364, 368 and others. “five nM” and “ten nM” should be “5 nM” and “10 nM”.

Lines 210-221. The text does not appear formatted correctly.

Lines 318, 321, 325, 399, 400, 419, 439, 440, 501, 502 and others. Please indicate city and state of production for the instruments and reagents used.

Line 330. Please replace “each genotype’s diquat resistance” with “diquat resistance of each genotype”.

Line 348. Please replace “aim” with “aimed”.

Lines 440-441. Why is “Macherey Nagel” written in capital letters?

Lines 483-485. “one hour, two hours, four hours…” and others should be “1 hour, 2 hours, 4 hours..” and so on.

Lines 507-510. Why are the parameters capitalized?

Reviewer 2 Report

Comments and Suggestions for Authors

This manuscript describes the brief mechanism of non- target-site resistance in duckweed. This manuscript well written and systematically describes the molecular mechanisms. We appreciate  author method to determine the systemic toxicity and GWAS analysis.This manuscript can be accepted with minor revision. My comments are as follow;

1. How this study varies in intra-species variations of different species .

2. Is there any significant difference between Figure 5A, C, D, E and F. 

Comments on the Quality of English Language

moderate English editing is required 
